# Fermi: Monitoring the Gamma-Ray Universe

**David J. Thompson**

NASA Goddard Space Flight Center, Greenbelt, MD 20771, USA; david.j.thompson@nasa.gov

**Abstract:** Since 2008, the Large Area Telescope and the Gamma-ray Burst Monitor on the *Fermi Gamma-ray Space Telescope* have been monitoring the entire sky at energies from about 8 keV to more than 1 TeV. Photon-level data and high-level data products are made publicly available in near-real time, and efforts continue to improve the response time. This long-duration, all-sky monitoring has enabled a broad range of science, from atmospheric phenomena on Earth to signals from high-redshift sources. The *Fermi* instrument teams have worked closely with multiwavelength and multi-messenger observers and theorists to maximize the scientific return from the observatory, and they look forward to continued cooperative efforts as *Fermi* moves into its second decade of operation.

**Keywords:** gamma rays; monitoring; variability

---

## 1. Introduction

Gamma rays, by definition the most energetic form of electromagnetic radiation, trace extreme, non-thermal processes taking place in the Universe. Monitoring of the $\gamma$-ray sky has shown that many $\gamma$-ray phenomena are highly variable, with time scales ranging from a fraction of a second to decades. Such variability offers opportunities for not only multiwavelength, but also multi-messenger studies, since $\gamma$ rays can be produced along with cosmic rays, neutrinos, and gravitational radiation.

Because $\gamma$ rays do not penetrate Earth's atmosphere, they must be studied with detectors outside the atmosphere or by using the atmosphere itself for indirect detection. The *Fermi Gamma-ray Space Telescope (Fermi)*, along with its smaller cousin *AGILE*, monitor the $\gamma$-ray sky from space at photon energies from about 8 keV to greater than 1 TeV. This review is focused on the monitoring programs of the *Fermi* mission and instruments: how they work, how results are shared, and some examples of the scientific return from ten years of operations.

## 2. Fermi Gamma-Ray Space Telescope

*Fermi* is a facility-class observatory built by an international team and operated by NASA and the U.S. Department of Energy [1]. By design, all *Fermi* $\gamma$-ray data become public immediately, along with analysis software and supporting documentation. The central repository for *Fermi*-related information is the *Fermi* Science Support Center (FSSC) at NASA Goddard Space Flight Center: https://fermi.gsfc.nasa.gov/ssc/. A list of abbreviations used in this review appears after Section 5. The two scientific instruments, the Large Area Telescope and the Gamma-ray Burst Monitor, are described below.

### 2.1. Large Area Telescope (LAT)

Operating at photon energies above about 20 MeV, the LAT detects $\gamma$ rays interacting by pair production to produce electrons and positrons [2]. The instrument produces arrival times, energies, and arrival directions for individual photons. Performance parameters of the LAT are summarized at http://www.slac.stanford.edu/exp/glast/groups/canda/lat$\_$Performance.htm. In terms of monitoring, a key parameter is the effective field of view, which is 2.4 steradians, allowing

instantaneous observation of nearly 20 percent of the sky. The principal limitation to monitoring the $\gamma$-ray sky with the LAT is counting statistics. With the exception of strong $\gamma$-ray bursts, even bright sources usually require exposures of tens of minutes to hours for a significant detection.

### 2.2. Gamma-Ray Burst Monitor (GBM)

As its name implies, the GBM is principally a sky monitor, using a set of 14 detectors on two sides of the spacecraft, pointed in different directions to observe nearly 8 steradians of the sky (the entire sky not occulted by Earth) instantaneously over the energy range 8 keV to 40 MeV [3]. Data include arrival times and energies for individual signals in each detector. Performance characteristics for the GBM are summarized at https://gammaray.nsstc.nasa.gov/gbm/instrument/. The GBM performance is constrained by instrumental background and is therefore most effective in identifying transient or periodic signals.

### 2.3. Fermi Operations

Taking advantage of the huge fields of view of the two scientific instruments, the *Fermi* operations team has operated the satellite largely in a scanning mode since its launch in June, 2008 [1]. By rocking the pointing direction north and south of the orbital plane on alternating 96-min orbits, the LAT and GBM survey the entire sky every $\sim$3 h. *Fermi* does have flexible pointing capability, and has sometimes been operated in modes that increase the exposure to individual targets or regions of the sky. Most such special operations have been carried out in response to Target of Opportunity requests, which can be made by anyone.

Because *Fermi* was planned as an extended mission, the spacecraft and instruments were built with flexibility in mind. The spacecraft, for example, has a propulsion system intended for controlled end-of-life de-orbiting. This system has been used (once) to maneuver *Fermi* away from a potential collision with a defunct spacecraft (see https://www.nasa.gov/mission$\_$$pages/GLAST/news/bullet-dodge.html). This capability remains in case of future near-miss situations. The operation of the GBM originally transmitted time-tagged event data only when an on-board trigger enabled this mode, but a change in 2012 switched to continuous time-tagged event data, allowing more sensitive searches for short $\gamma$-ray bursts [4]. Based on flight experience, the LAT team developed a new analysis process, Pass 8, that increased the rate of detected $\gamma$ rays by about 30 percent and with improved angular resolution [5]. Neither instrument has suffered any uncorrectable deterioration. The result is that after 10 years in space, the scientific capabilities of the *Fermi* instruments are better than at the time of launch. The instrument and operations teams are continuing to search for further improvements in performance.

In March 2018, the *Fermi* spacecraft experienced a failure: one of the two solar panels became unable to rotate to track the Sun. Although the solar array is still able to provide full power to the instruments and spacecraft, this limitation has necessitated a partial change of observing strategy. Instead of viewing the full sky every three hours most of the time, the LAT sometimes takes several weeks to produce an all-sky image. The trade-off is that the LAT obtains deeper short-term exposures of the part of the sky that it is viewing. The GBM is unaffected by this change of observing mode. The operations team continues to investigate ways to optimize the viewing plan.

## 3. Sharing *Fermi* Results

Although a sky-monitoring program produces a variety of scientific results that appear through standard publication channels, its impact is substantially increased by sharing of results to enable multiwavelength and multi-messenger studies. Making the data and software public is a first step, but most scientists outside the $\gamma$-ray community have limited interest in learning the data analysis systems. The *Fermi* instrument teams and others have therefore undertaken a variety of efforts to share higher-level results, using traditional, online, and rapid forms of communication.

## 3.1. Catalogs

A catalog is a very traditional way of sharing results of a sky survey. The *Fermi* instrument teams have published a number of catalogs, both general and specialized. The most recent versions of some of these are:

- Third Fermi GBM Gamma-Ray Burst Catalog [6]. This catalog covers the first six years of the mission and presents GBM results from 1405 $\gamma$-ray bursts (GRBs).
- Third Fermi LAT Catalog [7]. The 3FGL catalog includes LAT results from the first four years of the mission, with 3033 $\gamma$-ray sources.
- GBM Time-resolved Spectral Catalog of GRBs [8]. Using the first four years of GBM data, this catalog presents time-resolved spectra of 81 GRBs.
- GBM Magnetar Catalog [9]. This catalog presents five years of GBM data on flares from high-magnetic-field neutron stars (magnetars).
- GBM X-ray Burst Catalog [10]. In its first three years, GBM detected 1084 X-ray bursts.
- Third Fermi LAT Hard Sources Catalog [11]. This LAT catalog of sources seen at energies above 10 GeV contains 1556 objects.
- GBM Catalog of Terrestrial Gamma-ray Flashes [12]. Over 4000 terrestrial $\gamma$-ray flashes (short bursts associated with thunderstorms) are reported in this catalog.
- Second LAT Catalog of Flaring Sources [13]. The Fermi All-sky Variability Analysis (FAVA) monitors the high-energy sky for flaring activity, using an aperture photometry method to compare the $\gamma$-ray sky at any time with the average sky over many years. In over seven years of observation, FAVA found 4547 such flares.
- LAT Supernova Remnant Catalog [14]. This catalog presents results for GeV emission from 30 supernova remnants.
- Third LAT Catalog of Active Galactic Nuclei [15]. The 3LAC catalog is a supplement to the 3FGL catalog, focused on the properties of the 1591 active galactic nuclei (AGN) seen at high Galactic latitudes.
- LAT Gamma-ray Burst Catalog [16]. In three years of observations, the LAT detected 35 GRBs, whose properties are described in this catalog.
- Second LAT Pulsar Catalog [17]. Properties of pulsed $\gamma$ rays from 117 pulsars are summarized in this LAT catalog.

As the *Fermi* mission continues, updated versions of all these catalogs are expected. As of this writing, the 4FGL catalog is in preparation.

## 3.2. Online Resources

In today's electronic world, catalogs like those described in the previous section are most useful in online format. Many of these catalogs are available in interactive form at the FSSC data products site (https://fermi.gsfc.nasa.gov/ssc/data/access/). A mirror site at the Italian Space Agency's Space Science Data Center (http://www.asdc.asi.it) hosts copies of most of these as well as other useful online resources. For several of these data sources, the online catalogs are updated regularly, making them more current than the static published versions. One example is the FAVA analysis (https://fermi.gsfc.nasa.gov/ssc/data/access/lat/FAVA/), which is updated weekly, showing current flaring activity in the LAT data.

Electronic resources also include material not published in traditional forms. Here are some examples:

- Monitored Source List: https://fermi.gsfc.nasa.gov/ssc/data/access/lat/msl$_$lc/. A LAT automated analysis generates flux values for all sources whose daily $\gamma$-ray flux has exceeded $1 \times 10^{-6}$ photons (E > 100 MeV) cm$^{-2}$ s$^{-1}$ at least once since the start of the *Fermi* mission. Although these are not publication-quality light curves, they provide useful monitors of past and current activity for these sources. Daily and weekly light curves are updated regularly. An example is shown in Figure 1.

- Public List of LAT-Detected Gamma-ray Pulsars: https://confluence.slac.stanford.edu/x/5Jl6Bg. Newly detected $\gamma$-ray pulsars are added to this list regularly. This page includes details and references for each of more than 200 known pulsars.
- GBM Accreting Pulsar Histories: https://gammaray.nsstc.nasa.gov/gbm/science/pulsars.html. Over 30 accreting pulsars are monitored by the GBM in the energy range 12–50 keV. The pulse periods and flux values are updated regularly. An example is shown in Figure 2.
- LAT Aperture Photometry Light Curves: https://fermi.gsfc.nasa.gov/ssc/data/access/lat/4yr$_ $catalog/ap$_$lcs.php. Weekly updates of aperture photometry 30-day interval light curves for all 3FGL sources are maintained, along with periodicity analyses. An example is shown in Figure 3.
- Fermi Solar Flare X-ray and Gamma-ray Observations: https://hesperia.gsfc.nasa.gov/fermi$_ $solar/. This collection of links offers extensive information about both GBM and LAT solar observations.
- Very Important Project (VIP) list of AGN. The LAT team has identified 30 candidate AGN for concentrated study in the future: https://confluence.slac.stanford.edu/display/GLAMCOG/ VIP+List+of+AGNs+for+Continued+Study. For each one, LAT scientists have been identified who will be watching these sources, encouraging multiwavelength studies, and possibly even organizing full multiwavelength campaigns. These are ones for which the team will make a particular effort to cooperate with anyone interested in the same sources.

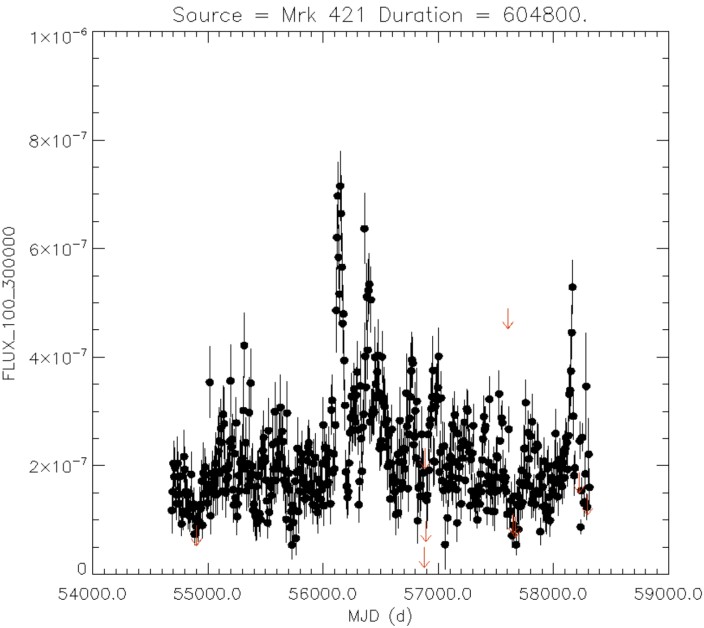

**Figure 1.** Weekly automated *Fermi*-LAT flux values (photons (100 MeV < E < 300,000 MeV) cm$^{-2}$ s$^{-1}$) for Mrk 421 as a function of Modified Julian Day (MJD).

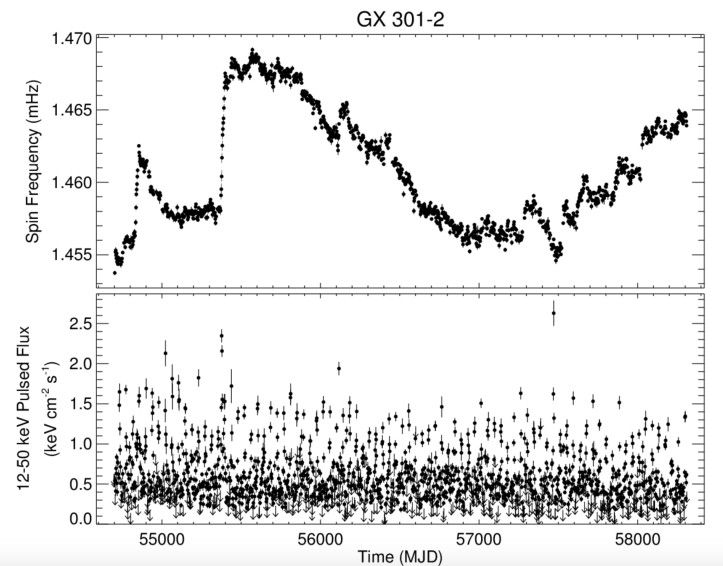

**Figure 2.** *Fermi*-GBM pulse periods and flux values for GX301−2 as a function of MJD.

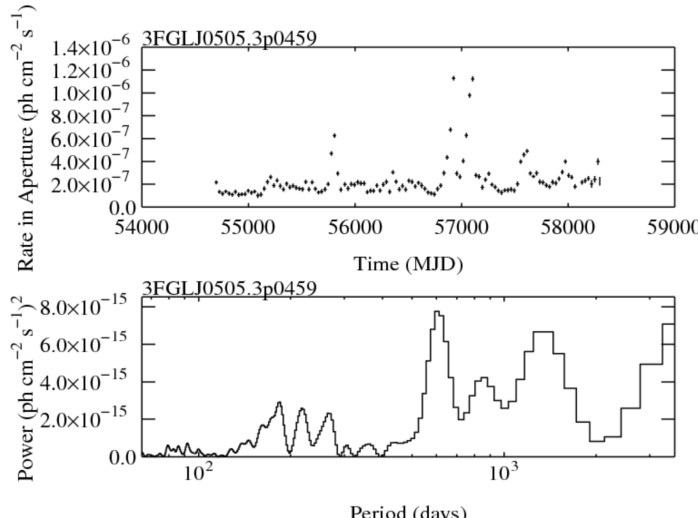

**Figure 3.** *Fermi*-LAT aperture photometry light curve and periodicity analysis for PKS 0502+049 as a function of MJD. The analysis shows no significant periodicity.

### 3.3. Rapid Communications

The nonthermal universe is highly variable, including transient, periodic, and randomly changing phenomena. Wide-field monitoring facilities like *Fermi* are ideal for observing such variability, but much of the electromagnetic spectrum is observed primarily by narrow-field instruments. Rapid communication of interesting events is therefore critical to maximizing the scientific return from the world-wide array of observing facilities. In addition to the routine updates of information described in the previous section, the *Fermi* instrument teams employ a variety of methods to disseminate alerts. These rapid communications depend on the intensity and duration of the $\gamma$ radiation, the amount of processing needed to extract the signal, and the appropriate audience to be informed. These communications methods are described below.

### 3.3.1. Public Communications

Most alerts from the *Fermi* instrument teams are broadcast to the public. These include, in approximate time order following the event:

- GRB on-board alerts sent through the Gamma-Ray Coordinates Network (GCN): https://gcn.gsfc.nasa.gov. GCN, also named Transient Astronomy Network (TAN), is a near-real-time system for rapid communications. GRBs are the only transients bright enough to be detected by the *Fermi* GBM or LAT on board the spacecraft. An Initial GCN Notice can be sent electronically within seconds of the GRB, followed shortly by a notice in a format that can be used to point automated telescopes and then about 15 min later by one or more with additional information about the burst. An archive of *Fermi* GCN notices can be found at https://gcn.gsfc.nasa.gov/fermi$_$grbs.html.
- GRB or other transient (such as high-energy neutrino events) follow-up analysis results sent as GCN Circulars. GCN Circulars are sent with more information about a transient, including the light curve shape and the spectrum. They are sent hours after the actual burst. An archive of GCN Circulars from many instruments can be found at https://gcn.gsfc.nasa.gov/gcn3$_$archive.html.
- Flaring activity from LAT sources sent as GCN Notices. LAT non-GRB flare detections are limited by counting statistics and therefore depend on accumulating data for analysis on the ground. The basic latency from the time of data collection at the instrument to having useful $\gamma$-ray information is about 6 h, including transmission of data in several steps and two levels of processing to extract the photon information. First analysis of the $\gamma$-ray data is done by the LAT team's Automated Science Processing system [18] working on daily data sets. A source that shows a daily flux with $5\sigma$ excess compared to the average flux over the preceding two weeks generates a GCN Notice. An archive of these notices can be found at https://gcn.gsfc.nasa.gov/fermi$_$lat$_$mon$_t$rans.html.
- Astronomer's Telegrams (ATels) : http://www.astronomerstelegram.org. ATels have become a standard means of rapid communication for a wide variety of astronomical phenomena. They are prepared and submitted manually, generally about a day after the actual event in the case of *Fermi*. For LAT results, ATels generally come from the volunteer Flare Advocates on the LAT science team [19].

Alerts such as these enable Target of Opportunity requests not only to *Fermi* but also to a wide range of other astronomical facilities. Close cooperation of astrophysicists from across the electromagnetic spectrum and multi-messenger facilities is increasingly a standard way to maximize the scientific return from all these resources.

### 3.3.2. Communications with Limited Distribution

Although the *Fermi* $\gamma$-ray data are all immediately public, there are occasions when the instrument teams do not broadcast results to the world but rather to a smaller audience. These exchanges tend to fall into two overlapping situations: (1) Some specialized results are primarily of interest to a limited number of scientists; and (2) Communication partners sometimes have proprietary resources whose uses need to remain confidential.

An example of the first case is the gammamw mailing list: https://lists.nasa.gov/mailman/listinfo/gammamw. This mailing list is moderated, and messages are archived. Membership in the list is open to anyone who is interested in multiwavelength studies of $\gamma$-ray sources, and any member can post messages. This list has been used primarily by the LAT team to inform the multiwavelength community of activity in the $\gamma$-ray sky other than those events dramatic enough to warrant an ATel, but it has also been used to announce data availability from other wavelengths of possible interest to the $\gamma$-ray community.

The second case includes exchanges of information with various facilities carried out under Memoranda of Understanding (MOUs). Such information exchanges can take two paths:

- Limited-access GCN Notices and Circulars have been used to share information between the GBM team and the LIGO/Virgo gravitational wave observatories [20,21]. These MOU-based exchanges were of sub-threshold events seen by each facility, with the idea that seeing something simultaneously could increase the chances of finding a real signal.

- Informal sharing of information, usually by e-mail, has been undertaken between *Fermi* instrument teams and a wide variety of multiwavelength and multi-messenger facilities. A principal use of this channel has been alerts from the LAT team to the narrow-field imaging atmospheric Cherenkov telescopes (IACTs): H.E.S.S., VERITAS, MAGIC, and FACT. Although the LAT has limited sensitivity at energies that overlap with the IACTs, even a few source photons seen by LAT with energies above 10 GeV can suggest that an observation at TeV energies might be worthwhile. The LAT team also participates in the MAGIC/HESS/VERITAS/HAWC/FACT flaring-AGN agreement, another e-mail-based information exchange.

Because the *Fermi* data become public at the same time they are available to the instrument teams, rapid communications of *Fermi* results from external observers are also possible. Examples of techniques used for this purpose are [22,23]. ATels have been published based on [23], e.g., http://www.astronomerstelegram.org/?read=11874.

## 4. Some Examples of Scientific Results from Monitoring

Essentially all *Fermi* results come from its monitoring of the $\gamma$-ray sky. A summary of all these results is far beyond the scope of this review. Instead, some examples illustrate the various ways monitoring can translate into scientific results.

### 4.1. Monitoring to Accumulate Statistics

For a statistics-limited instrument like the *Fermi* LAT, monitoring the sky opens discovery space for new sources thanks to deeper exposures. Here are two examples:

- The 2FHL Catalog. With 80 months of LAT data, it was possible to construct the 2FHL catalog of sources in the energy range 50 GeV–2 TeV [24]. These 360 sources, shown in Figure 4, include some that were seen with as few as 3 photons. Only about 25 percent of these sources had been detected by IACTs, making this catalog a valuable roadmap for future TeV observatories like the Cherenkov Telescope Array (CTA).
- A Radio-quiet Millisecond Pulsar. Using 5.5 years of LAT data, combined with advanced pulse-searching methods, the Einstein@Home distributed computing project was able to reveal two millisecond pulsars in blind searches, and one of these was the first radio-quiet millisecond pulsar [25]. Although the Einstein@Home project was able to accumulate the equivalent of 10,000 years of CPU time, this computing power still required enough detected $\gamma$ rays to make this discovery. The pulse profiles of the two pulsars, along with the pulse phases of individual $\gamma$ rays, are shown in Figure 5.

### 4.2. Monitoring to Search for Long-Term Trends

The *Fermi* monitoring of the full sky has provided long-term, consistent data sets for $\gamma$-ray sources, enhancing time-series analyses. These data offer opportunities to search for trends that might not be visible on shorter time scales. Here are three examples:

- Accreting Pulsars. As mentioned above, the GBM monitors a number of accreting pulsars: https://gammaray.nsstc.nasa.gov/gbm/science/pulsars.html. Unlike the example in Figure 2, some other pulsars in this set do show clear long-term trends: 4U 1626−67 has been spinning up since the beginning of the *Fermi* mission (https://gammaray.nsstc.nasa.gov/gbm/science/pulsars/lightcurves/4u1626.html), while the pulse frequency of GX 1 + 4 has been decreasing steadily (https://gammaray.nsstc.nasa.gov/gbm/science/pulsars/lightcurves/gx1p4.html).
- Possible AGN Periodicity. BL Lac object PG 1553 + 113 is bright enough in the LAT data to be detected regularly, and the long-term light curve showed evidence of quasi-periodic 2.2-year variations in the $\gamma$-ray flux [26]. This periodicity, now reaching 5 cycles and shown in Figure 6, suggests that this system may contain two supermassive black holes [27].

- Power Spectral Density Analysis. Characterization of variability by power density spectra can show features on various time scales, such as the example of LAT analysis for blazar OJ 287 [28].

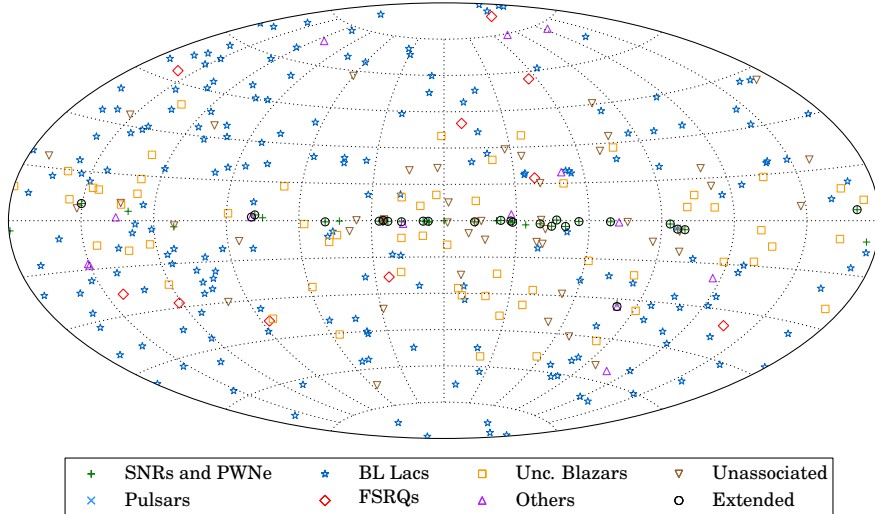

**Figure 4.** Sky map, in Galactic coordinates and Hammer-Aitoff projection, showing the sources in the 2FHL catalog classified by their most likely association [24]. ( ©American Astronomical Society, AAS. Reproduced with permission.)

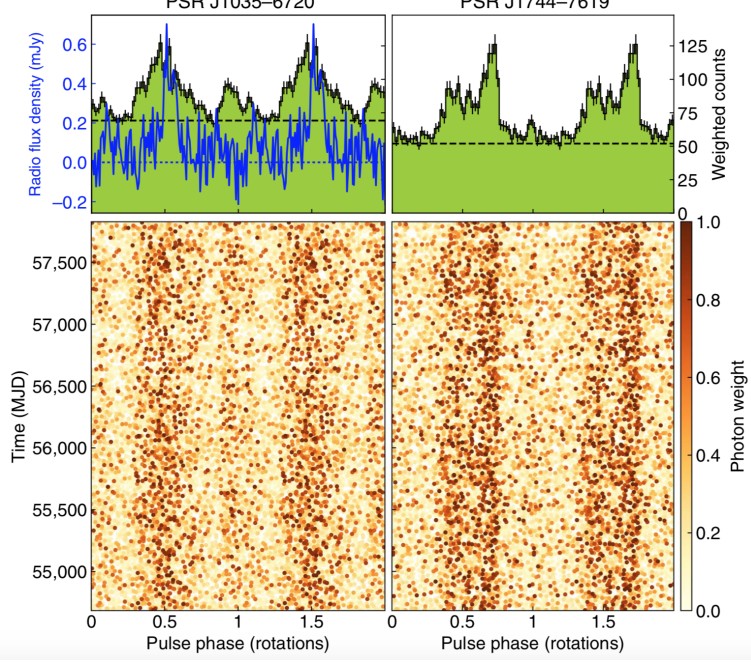

**Figure 5.** Rotational phases of individual $\gamma$ rays (**bottom**) and integrated pulse profiles (**top**) of the newly detected millisecond pulsars. Each photon has been assigned a weight, determined by its energy and arrival direction, representing the probability of it having come from the $\gamma$-ray source in question. The blue line is the radio signal from PSR J1035$-$6720 [25]. (©American Association for the Advancement of Science. Reproduced with permission.)

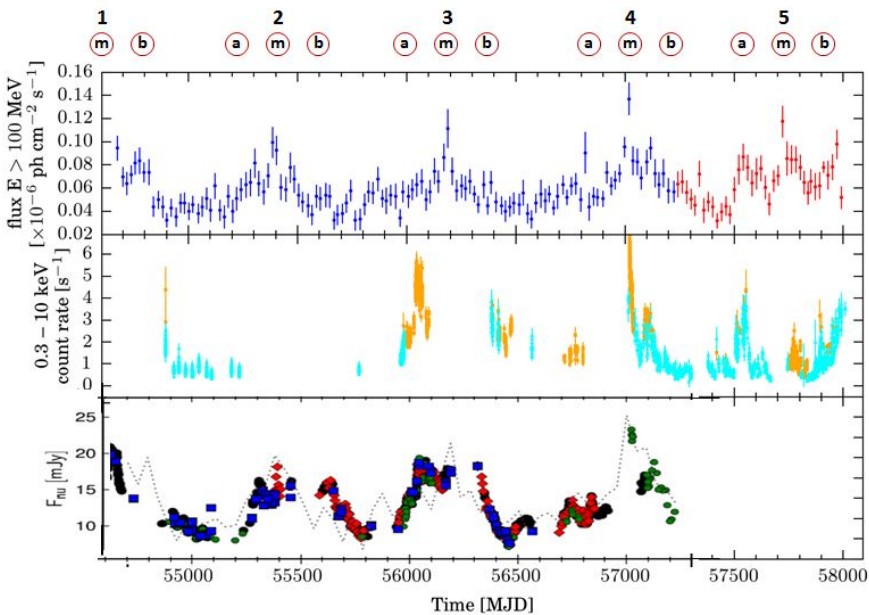

**Figure 6.** Light curves of PG 1553 + 113 in various bands [27]. **Top** panel: $\gamma$ rays from *Fermi* LAT, with the 2016–2017 data traced in red. **Middle** panel: keV X-rays from *Swift*. **Bottom** panel: optical R band reported from Figure 2 of [26]. (©AAS. Reproduced with permission.)

*4.3. Monitoring to Detect Rare Transients*

For variable or transient phenomena, which are often unpredictable, extreme cases can provide the greatest insight. Long-term, wide-field monitoring increases the chances of seeing such rare events. Three *Fermi* examples are:

- GRB 090510. This short, bright, distant (z = 0.9) GRB was seen by both the GBM and the LAT, with energies up to 31 GeV. The near-simultaneous arrival times of photons with a broad range of energies set strong constraints on Lorentz-invariance violation and some models of quantum gravity [29].
- Crab Flares. GeV flares from the Crab were discovered by *AGILE* and *Fermi* LAT [30,31]. Rapid variability of such flares seen by LAT (time scale of $\sim$5 h) suggests particle acceleration processes other than shock acceleration [32].
- Shock-powered Nova. ASASSN$-$16 ma, appearing two years ago, is one of the brightest $\gamma$-ray novae yet seen (Figure 7). The excellent optical and $\gamma$-ray coverage showed the light curves to match, implying that both bands were powered by shocks, contrary to expectation that nova optical emission was related to the white dwarf in these binary systems [33].

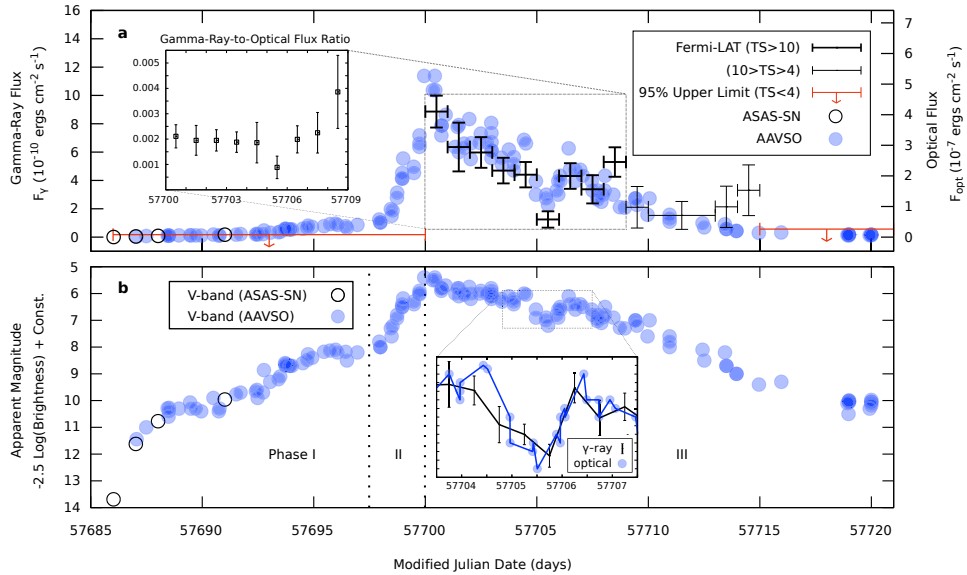

**Figure 7.** The top panel (**a**) shows the $\gamma$-ray (black and gray crosses, and red arrows) and bolometric (blue and black circles) light curves of ASASSN-16ma in flux units of ergs cm$^{-2}$ s$^{-1}$, using observations from *Fermi* LAT and ASAS-SN/AAVSO. The bottom panel (**b**) shows the V-band light curves of the same optical datasets on a magnitude (logarithmic) scale. The inset box in (**b**) zooms in on the emission dip at MJD 57705 in $\gamma$ rays and optical, which directly show the co-variance of the $\gamma$-ray and optical emission on time-scales as short as 0.5 days [33]. (©Springer Nature. Reproduced with permission.)

By continuing to monitor the sky, the *Fermi* instruments are well-prepared for the next extreme transient.

### 4.4. Monitoring for Synergy with New Facilities

Resources available to study the nonthermal universe have changed since the launch of *Fermi*, and that has presented new opportunities for discoveries that depend on two or more facilities, i.e., multiwavelength or multi-messenger results. Here are two recent examples:

- GRB 170817A/GW170817. The first announcement of this event, a binary star merger that produced a gravitational wave signal, came from a GCN Notice of an on-board *Fermi*-GBM trigger, followed by a GCN Notice from the LIGO-Virgo gravitational wave teams and then a flood of other observations [34] (see Figure 8). Among the many results from this event, two that involved *Fermi* were the confirmation of neutron star mergers as the origin of short GRBs and the first measurement of the speed of gravity as essentially the speed of light.
- Swift J0243.6+6124. In 2017 September, a massive outburst from a newly discovered Be/X-ray binary system was detected by the *Swift* Burst Alert Telescope [35], making it the first known Galactic ultraluminous X-ray pulsar [36]. The full nature of this unusual system, Swift J0243.6+6124, was only revealed by combining data from *Swift*, *Fermi*-GBM, *NuSTAR*, *NICER*, and *Gaia*, with the GBM providing the detailed evolution of the flux and pulse spin frequency (Figure 9). Of the five missions involved, only *Swift* was in operation at the time *Fermi* was launched.

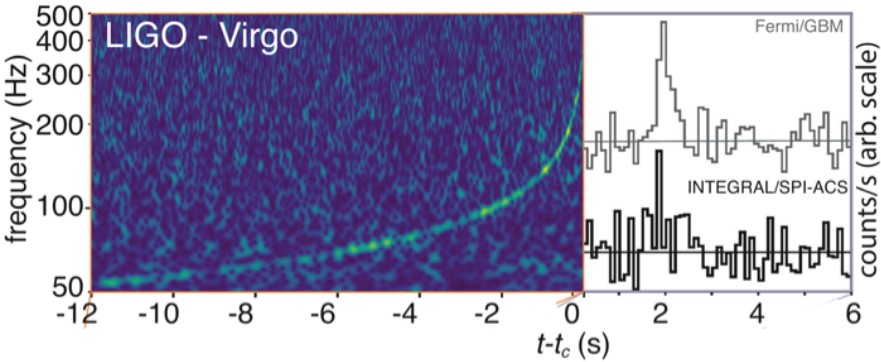

**Figure 8.** Timeline of the GRB 170817A/GW170817 event [34]. (©AAS. Reproduced with permission.)

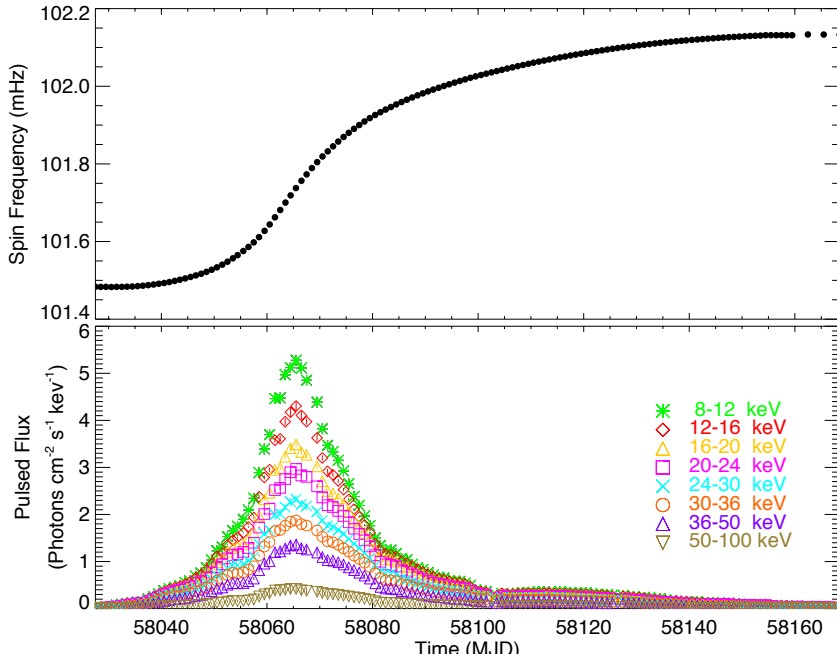

**Figure 9.** (**Top**): Barycentered and orbit-corrected spin-frequency history of Swift J0243.6+6124 measured with GBM. (**Bottom**): Pulsed flux measured with GBM in nine energy bands [36]. (©AAS. Reproduced with permission.)

### 4.5. Monitoring to Establish Context

In some cases, an astrophysical observation is only meaningful in the context of historical or comparative observations. An example is the coincidence of an IceCube high-energy neutrino with a flaring $\gamma$-ray blazar. In 2017 September, the IceCube Collaboration released a GCN Notice (through the Astrophysical Multimessenger Observatory Network, AMON. See https://www.amon.psu.edu) about a neutrino designated IceCube-170922A [37]. *Fermi*-LAT data showed that the neutrino came from a direction positionally coincident with a known $\gamma$-ray blazar, TXS 0506+056, which was in a flaring state [38]. The localization is shown in Figure 10. Because IceCube and *Fermi* are both monitoring facilities, it was possible to construct an empirical calculation of the probability that this was a chance coincidence, based on the history of IceCube alerts and a comparison with over 2000 *Fermi*-LAT blazar light curves [39]. The resulting $\sim 3\sigma$ significance did not depend on detailed modeling or multiple assumptions.

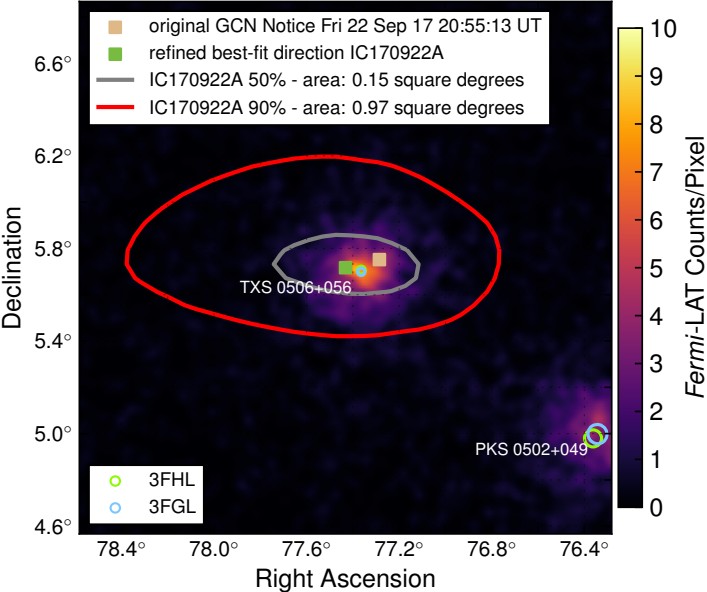

**Figure 10.** Sky position of IceCube-170922A overlaying the $\gamma$-ray map from *Fermi* LAT above 1 GeV [39]. The tan square indicates the position reported in the initial alert and the green square indicates the final best-fitting position from follow-up reconstructions [37]. Gray and red curves show the 50% and 90% neutrino containment regions, respectively, including statistical and systematic errors. *Fermi*-LAT data are shown as a photon counts map in 9.5 years of data in units of counts per pixel, using detected photons with energy of 1 to 300 GeV in a 2° by 2° region around TXS 0506+056. Also shown are the locations of a $\gamma$-ray source observed by *Fermi*-LAT as given in the 3FGL [7] and 3FHL [11] source catalogs, including TXS 0506+056. For *Fermi*-LAT catalog objects, marker sizes indicate the 95% confidence level positional uncertainty of the source. (©American Association for the Advancement of Science. Reproduced with permission.)

## 5. Discussion

Modern communications, i.e., the Internet, and high-speed computing have enabled a dramatic improvement in cooperative multiwavelength and multi-messenger exploration of the universe. Monitoring missions like *Fermi* have contributed to this broadband approach to astrophysics, in particular by sharing data and high-level results rapidly with the community. As new resources become available, the *Fermi* instrument and operations teams will be seeking new avenues of cooperation. The *Fermi* instruments are performing well; the *Fermi* orbit is stable for decades to come; there are no expendable supplies on *Fermi* to run out; and the flexibility of operations offers assurance that *Fermi* $\gamma$-ray data can be available for many years to come.

**Acknowledgments:** This summary was made possible by the work of hundreds of contributors to the *Fermi* instruments and mission, along with the many multiwavelength and multi-messenger scientists whose efforts complement the $\gamma$-ray studies.

**Funding:** This research received no external funding.

**Conflicts of Interest:** The author declares no conflict of interest.

## Abbreviations

The following abbreviations are used in this manuscript:

| | |
|---|---|
| AGN | Active Galactic Nucleus |
| ATel | Astronomer's Telegram |
| CTA | Cherenkov Telescope Array |
| GBM | Gamma-ray Burst Monitor |

GCN        Gamma-ray Coordinates Network
GRB        Gamma-ray Burst
FAVA       Fermi All-sky Variability Analysis
FSSC       Fermi Science Support Center
IACT       Imaging Atmospheric Cherenkov Telescope
LAT        Large Area Telescope
LIGO       Laser Interferometer Gravitational-wave Observatory
MJD        Modified Julian Day
MOU        Memorandum of Understanding
3FGL       Third Fermi LAT Catalog

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
