# Peer review of "Fermi: Monitoring the Gamma-Ray Universe"

_galaxies, doi:10.3390/galaxies6040117_

Round 1

Reviewer 1 Report

Review of Fermi: Monitoring the Gamma-Ray Universe

Author- David J. Thompson

Although the paper does not present new data or new science, it is a very good review article which might stimulate new research in the future.  It is well written, easy to read, and is filled with web sites that researchers like myself may not be familiar with, and might potentially yield new results coupled with our own observations.  It I in this mode that I think the paper should be published.

The Introduction states that “…photon energies from less than 10 Kev to greater than 1 TeV.”  This is very vague.  It would be more informative to state either more accuracy, or error bars on the limits. 

I checked all of the links and they work well.

Captions on the Figure 1 needs some work. Figure 1 “Flux….100….300000” I assume means form 1000 MEV to 30000 Mev, but the caption should read Photons/cm-2/s-1.

Figure 5 needs further explanation in the text.  It appears to be very interesting, but not clear what it really means.  Further explanation would be great here.

It would be nice to have the abbreviations table earlier in the text, or maybe just tell the reader where to find the abbreviations early on.  I read the paper wondering what most of the abbreviations meant, then finally found the key and had to go back and associate what I had read with the appropriate instrument or catalog.

new

Author Response

My thanks to the reviewer for these comments.  Responses to the specific comments are given below. 

The Introduction states that “…photon energies from less than 10 Kev to greater than 1 TeV.”  This is very vague.  It would be more informative to state either more accuracy, or error bars on the limits.  

=> I added the specific lower limit for the GBM (8 keV).  There is no well-defined upper limit to the LAT energy range; therefore I left the text unchanged.

Captions on the Figure 1 needs some work. Figure 1 “Flux….100….300000” I assume means form 1000 MEV to 30000 Mev, but the caption should read Photons/cm-2/s-1.

=>  I added detail to the figure caption as requested. 

Figure 5 needs further explanation in the text.  It appears to be very interesting, but not clear what it really means.  Further explanation would be great here.

=> I added a sentence in the text describing the figure. 

It would be nice to have the abbreviations table earlier in the text, or maybe just tell the reader where to find the abbreviations early on.  I read the paper wondering what most of the abbreviations meant, then finally found the key and had to go back and associate what I had read with the appropriate instrument or catalog.

=> I added a sentence in section 2 referring to the list of abbreviations. 

Reviewer 2 Report

While the authors do indeed mention ToO observations and other multifrequency collaborations, perhaps another sentence or two about these are in order.

I would urge, if you have not already done so, that more time series analysis (besides just looking for periodicities) be made of this huge amount of data taken over the years.  If indeed, you have done more, it should be mentioned in the paper.  If not, I would urge you to be more extensive in your analysis.

You have done an excellent job in doing and explaining the synergy between your observations and other instruments, the internet, and outside collaborations in general.  You have made maximum use of our most modern technology, communication, and use of the scientific community.

Author Response

My thanks to the reviewer.  Here are responses to the specific comments.

While the authors do indeed mention ToO observations and other multifrequency collaborations, perhaps another sentence or two about these are in order.

=> I added sentences in the Public Communications sections to emphasize these. 

I would urge, if you have not already done so, that more time series analysis (besides just looking for periodicities) be made of this huge amount of data taken over the years.  If indeed, you have done more, it should be mentioned in the paper.  If not, I would urge you to be more extensive in your analysis.

=> I added a third example, Power Spectral Density analysis, to illustrate that the Fermi data have other time-series applications than just periodicity. 

Reviewer 3 Report

Dear the Editor and the Authors,

This manuscript summarizes monitoring capabilities and results of the Fermi Gamma-ray Space Telescope. The overview is well summarized and informative for readers in the community. I would like to recommend this review article for publication in Galaxies. 

I have a few very minor comments as below.

- lines 68-73

How many weeks or months will it take to have an uniform exposure like the previous survey mode?

- Fig. 8

If possible, it is better to erase the orange and purple lines.

- Sec. 4.5

Ice Cube

->

IceCube

Author Response

My thanks to the reviewer for these comments.  My specific replies are below:

- lines 68-73

How many weeks or months will it take to have an uniform exposure like the previous survey mode?

=> The text says several weeks, and since optimization of the strategy is still being done, the Fermi team prefers not to make specific promises at this time. 

- Fig. 8

If possible, it is better to erase the orange and purple lines.

=> Most of the lines are now erased, but since I do not have the original graphic, I was not able to erase them completely. 

- Sec. 4.5

Ice Cube

->

IceCube

=> Changes made.